# Dynamics of Major and Trace Elements in Water–Soil–Tree Interaction: Translocation in *Pyrus malus* in Chihuahua, Mexico Using ICP-OES and Its Health Risk Implications

**DOI:** 10.3390/ijerph191912032

**Published:** 2022-09-23

**Authors:** Angélica Cervantes-Trejo, Luz O. Leal

**Affiliations:** 1Tecnológico Nacional de México Campus Chihuahua, Avenida Tecnológico 2909, Chihuahua 31310, Mexico; 2Environment and Energy Department, Advanced Materials Research Center (CIMAV) S.C., Miguel de Cervantes 120, Chihuahua 31109, Mexico

**Keywords:** trace elements, Hazard Quotient, carcinogenic risk assessment, health risk, apple, ICP-OES, water–soil–plant interaction, accumulation in plants

## Abstract

The transference of metals from water irrigation and soil to plants is a possible pathway of contamination for the trophic chain. This research is focused on the distribution of 16 analytes in the water–soil–tree (*Pyrus malus*) interaction in an agricultural region in the state of Chihuahua in Mexico from August 2019 (first sampling) to August 2020 (second sampling). The apple variety under investigation was Golden Delicious; it was found that the trace elements of As (0.18–0.34 mg·kg^−1^) and Cd (0.11–0.14 mg·kg^−1^) in the apple were above the corresponding permissible limit, according to FAO/WHO, and Cr (0.08–0.86 mg·kg^−1^) was below the limit. Furthermore, the health risk implications were estimated by the Hazard Quotients (HQ) and carcinogenic risk (CR). For carcinogenic risk, As, Cd, and Cr exceeded the risk limit (CR > 10^−4^). This investigation as well provides a link for similar research around the globe. Major and trace elements detection was performed with the Inductively Coupled Plasma-Optical Emission Spectrometry (ICP-OES) technique, along with a prior homogenization of samples and microwave acid digestion. To obtain the statistical behavior, an analysis of variance and correlation was performed.

## 1. Introduction

Urbanization, industrialization, farming, and pathways of transference are responsible for the global increase in trace elements in irrigation water and agricultural soil. Nevertheless, there is a lack of research on the risk and health implications of the bioaccumulation of trace elements in living beings. Plants can bioaccumulate trace elements from contaminated water and soil [1,2,3]. The major and trace element concentrations in plants are known to be affected by the variety of the crop, soil conditions, weather conditions during the plant’s growth, use of fertilizers, and the state of the maturity of the harvest [4,5]. The trace elements (TEs) contamination in irrigation water and agricultural soils has drawn worldwide attention for its health implications by consumption of the food produced under these conditions [6,7,8,9,10] because these elements have the following characteristics: they are non-biodegradable, harmful to living organisms, carcinogenic, and also, they have accumulative and lasting capabilities. The arrangement of trace elements in the environment is conditioned by their local mineralization interactions and the weather-specific conditions of each site [11]. The use of fertilizers and pesticides, industrial emissions, and transportation resulted in a significant increase in the heavy metal content of the agricultural soil. In the same way, the plants consume both nutrients and toxic metals by absorbing them from contaminated irrigation water and agricultural soil as well as from air parts of the plants exposed to polluted environments [12,13]. These metals are first absorbed in the roots and then transported and translocated in different parts of the plant through various pathways. The processes of transport and absorption are different for every plant species. These phenomena depend on the different chemical differences and the characteristics of the environment, such as the pH, electrical conductivity, area of study, and site geology, among other processes [14,15].

According to several studies, people are exposed to TEs by ingestion; an important part of the nourishment diet consists of fruits and vegetables, which contain vitamins and minerals. The minerals are classified into micro (trace) or macro (major) elements. The macro-minerals include potassium (K), calcium (Ca), sodium (Na), magnesium (Mg), phosphorus (P), and sulfur (S), and micro-minerals (mainly metals) include iron (Fe), zinc (Zn), manganese (Mn), and copper (Cu). Trace elements have no known function in the plants; nevertheless, lead (Pb), cadmium (Cd), chromium (Cr), arsenic (As), antimony (Sb), and selenium (Se) can also be found in fruits and vegetables and those can be accumulated in the food chain [16,17,18]. Apples are well-known and widely consumed fruits of the *Pyrus malus* belonging to the Rosaceae family, which represent a good source of vitamins and minerals [19]. Studies are focused mainly on the polyphenolic composition and antioxidant activity of apples and to a lesser extent on the major and trace elements content, the transfer pathway, and the health risk. Therefore, studying the content of TEs in food and analyzing the carcinogenic and non-carcinogenic effects on human health is of high importance [20]. Therefore, it is necessary to analyze and assess the approximate levels of TEs in local fruits grown in the agricultural areas in Chihuahua. This research is important to understand the dynamics of trace elements in agricultural apple fields to know the health implications, through Hazard Quotients (HQ) and carcinogenic risk (CR).

The apple-producing region of Chihuahua is the national leader in apple production with an overall production of 561,000 tons, which represents 70% of the apples produced in Mexico, the main grown variety being Golden Delicious (60%). The apple production region is located to the northwest of Chihuahua state [21]. The goal of this research is to determine the dynamics of the major and trace elements in the water, soil, and plant interaction; this system is studied in the *Pyrus malus* tree with the variety Golden Delicious in four municipalities: Temósachic, Bachíniva, Namiquípa, and Cuauhtemoc Chihuahua. Finally, the health risk of the apples produced in the studied regions is evaluated through the Hazard Quotient (HQ) and the carcinogenic risk (CR); these indicators were estimated by the apple intake.

## 2. Materials and Methods

### 2.1. Study Area

In this research, six orchards across four counties in the state of Chihuahua were studied. These four counties are Temósachic, Bachíniva, Namiquípa, and Cuauhtemoc, which are municipalities in the state of Chihuahua, which is in the north of Mexico. In Figure 1, the area of study and the sampling points are presented. The agricultural area is in the west of the state, in the Sierra Madre Occidental; in this region, sub-humid conditions prevail with an annual temperature mean of 14 °C and an annual precipitation average of 445 mm in Temósachic, 409 mm in Bachíniva, 482 mm in Namiquípa, and 528 mm in Cuauhtémoc [22].

### 2.2. Sample Collection and Treatment

This study took place in six apple orchards: Rancho Verde and Cristina in Temósachic, La Esperanza in Bachíniva, Solis and La Mesa in Namiquípa, and Las Bebas in Cuauhtémoc. The sampling took place prior to the harvest of the apples in August to September 2019 for Rancho Verde, Cristina, and La Esperanza, and in August 2020 for Solis, La Mesa, and Las Bebas. Samples of irrigation water, soil, tree leaves, and apples were collected in every orchard, for an overall total of 162 samples in the two years of sampling (2019 and 2020). The samples of irrigation water were obtained from spring water in Rancho Verde and Cristina; in the cases of La Esperanza, Solis, and La Mesa, the water was pumped from wells; at last, the water used in Las Bebas was treated water. The water samples were collected in a 1-L bottles. In situ parameters, such as pH and electrical conductivity (EC), for water samples and soil were measured.

The sample collection method that was used was a systematic random sampling which considered 6 trees per orchard; each tree was considered a sampling unit, from which it was sampled also randomly 6 apples (edible fruit and peel), tree leaves, soil as close as possible to the tree, and irrigation water used in the orchard. Each tree was from a random point in the orchard to try to capture the variability across the orchard.

### 2.3. Digestion of Samples

An aliquot of soil (500 mg), leaves (200 mg), peels (250 mg), walnuts (250 mg), apples (250 mg), and irrigation water (45 mL) were taken and transferred to a poly vial of tetrafluoroethylene (PTFE). The samples were exposed to acid digestion by using different reagents. For soils, leaves, and water irrigation, 9 mL of HNO_3_ (69–70% J.T. Baker Instra-Analyzed) and 3 mL of HCl (36.5–38% J.T. Baker Instra-Analyzed) were added to the aliquots. The apples were peeled and homogenized. For peel and apple, 9 mL of HNO_3_ (69–70% J.T. Baker Instra-Analyzed) and 1 mL of H_2_O_2_ (30% J.T. Baker) were added to the aliquots. After tight closure, the vessels were placed into a microwave MARS Xpress (CEM). After acid digestion, the vessels were cooled at room temperature, and the samples were quantitatively recovered by filtration in 50-milileter class A volumetric flasks, then brought to 50 mL with Mili-Q water.

### 2.4. Inductively Coupled Plasma-Atomic Emission Spectrometry (ICP-OES) Conditions

All the chemical analyses of irrigation water, soil, leaves, apples, and peels were made by inductively coupled plasma-atomic emission spectrometry. The ICP-OES used was ICP-OES IcapTM 7000 series, manufactured by Thermo ScientificTM with an Ultrasonic Nebulizer CETAC U5000AT+. The use of an Ultrasonic Nebulizer in conjunction with an ICP-OES has long been accepted as a simple and cost-effective way to increase sensitivity and decrease detection limits.

The standard solutions were prepared from 1000 mg·L^−1^ for each element (National Institute of Standards and Technology, NIST; Gaithersburg, MD, USA). All measurements were performed in triplicates. The elements measured with the equipment were As, Ca, Cd, Cu, Cr, Fe, K, Mg, Mn, P, Pb, S, Sb, Se, and Zn.

### 2.5. Statistical Analysis

To identify the statistical behavior of variables obtained, an analysis of variance and correlation was performed. The procedure GLM (General Linear Model) in SAS 9 (Statistical Analysis System, 2002) (SAS institute, Cary, NC, USA) was used to obtain the estimators.

To determine the levels of variability, the association, and the behavior of absorption patterns between the elements, bivariate and multivariate analysis techniques were used. Initially, an analysis of correlations between pairs of variables of each matrix was analyzed for the trace elements and nutrients, using the CORR procedure of the package SAS (SAS 9.0). Subsequently, a cluster analysis was carried out with the physicochemical variables of irrigation water and soil (temperature, pH, electrical conductivity) and for the chemical variables, the concentration of nutrients and the trace elements in water, soil, leaves, apple, and peel. For the cluster analysis, the procedure CLUSTER from SAS software was used.

### 2.6. Estimated Daily Intake (EDI)

The estimated daily intake (EDI) was calculated with the following Equation (1) [23,24].
EDI = C × IR × EF × ED/(Bw × AT)(1)

In the numerator, C is the concentration mean of TE (mg/kg), IR is the ingestion rate of food (kg/day), EF is the exposure frequency (365 days/year), and ED is the exposure duration (70 years). In the denominator, BW refers to the average body weight (72 kg), and AT is the average time of the dose. For assessment of the carcinogenic risk, an AT of 70 years (25,550 days) was considered [25].

### 2.7. Health Risk Assessment

#### 2.7.1. Non-Carcinogenic Risk Assessment

The health risk assessment (non-carcinogenic and carcinogenic risk) of heavy metals in fruits was estimated via the model described by the EPA [26]. The Hazard Quotient (HQ) was used to calculate the non-carcinogenic risk of heavy metals (Equation (2)) [10].
HQ = EDI/RfD(2)
where RfD is the daily intake reference dose (mg/kg of body weight per day), an estimate of daily exposure to the human population that is likely to be without an appreciable risk of deleterious effects during a lifetime [24].

#### 2.7.2. Carcinogenic Risk Assessment

The carcinogenic risk assessment for trace elements was estimated according to the equation below (Equation (3)) [27]:CR = SF × EDI(3)
where CR is a carcinogenic risk for a lifetime, and SF is a slope factor for estimating the probability of an individual developing cancer from exposure to the contaminant for a lifetime. The SF for As, Cd, and Cr are 1.5, 0.38, and 0.5 (mg/kg/day^−1^), respectively [10]. Overall, CR < 10^−6^ is the safe limit, CR of 10^−6^ to 10^−4^ is the acceptable cancer risk range depending on the exposure circumstances, and CR > 10^−4^ is the threshold limit and is considered unacceptable [25].

## 3. Results

### 3.1. Chemical Parameters In Situ and Concentration of Major and Trace Elements in Soil and Irrigation Water

The parameters measured in soil and irrigation water samples collected in the study area are shown in Table 1.

The measured pH in the irrigation water tends to be neutral with a maximum value of 8.14, which is a slightly alkaline value [28]. However, the measured pH of the soils tends to be acid; for example, the orchard Las Bebas showed a pH value of 5.54, which is a strong acid value. Soils that have below 5.5 generally have low availability of calcium, magnesium, and phosphorus; because of those low values of pH, the solubility of iron is high [29].

The results of the ICP-OES analysis show the concentration of major and TEs in irrigation water (Table 2) of the six apple orchards. From the results, it can be observed that Sb was only detected in irrigation water collected from Las Bebas. From the results, it can be observed that Cd, Pb, and Se were not found in irrigation water.

The results of the concentrations of elements in the soil are presented in Table 3. As and Cd were not found in soils collected from Solis, La Mesa, and Las Bebas orchards. Likewise, it may observed that Sb and Se were not found in the soil.

### 3.2. Concentration of Major and Trace Elements in Pyrus malus

In Table 4, the averages of 14 element concentrations in apples and peel samples from six apple orchards in the apple-producing region of Chihuahua are presented. The results are the average concentration in peel and apple.

It may be observed that Pb and Se were not found in apples and peels. Likewise, Sb was not present in the apple; however, the Sb was only detected in peel collected from Solis, La Mesa, and Las Bebas orchards.

### 3.3. Concentration of Major and Trace Elements in Apple Leaves

The elements present in leaves were found in the following order of abundance (Table 5): Ca, K, P, Mg, and S > 1000 mg·kg^−1^; Na and Fe > 100 mg·kg^−1^; Zn and Mn > 10 Mg·kg^−1^; finally, Cu, Cr, As, Pb and Cd < 10 mg·kg^−1^.

The method was validated using Standard Reference Material (SRM) NIST 1515 apple leaves. For trace elements As, Cr, and major element S, which are not included in the list of certified values for elements in SRM, the tests were carried out by the addition of a standard solution. The recovery factors for the three added concentrations of As, Cr, and S varied from 98.23% to 106.32%, confirming the good precision of this method. Three replicates were analyzed for each sample, and the concentrations of the elements were evaluated as the mean of the three measurements. It received good repeatability (less than 10%) for all the measurements.

### 3.4. Non-Carcinogenic Risk Assessment

In Figure 2, the HQ values obtained for the most contaminated orchards are shown, presenting only the elements with high values for the non-carcinogenic risk assessment; these elements were As, Cd, Cr, and Zn. These values indicate the health risk assessment by chronic exposure to apple consumption, assuming ingestion of 200 g per day, which is an approximate weight of one apple. The element with the highest value of HQ was cadmium, followed by arsenic, zinc, and chromium.

### 3.5. Carcinogenic Risk Assessment

The CR values of As, Cd, and Cr were estimated and presented in Figure 3. The mean CR values of As (1.40 × 10^−3^), Cd (1.34 × 10^−4^), and Cr (9.58 × 10^−4^) exceeded the threshold limit, meaning that there is a potential carcinogenic risk for the long-term ingestion of the apples produced in these orchards (CR > 10^−4^) (Table 6).

## 4. Discussion

### 4.1. Element Characterization in Water and Soil

Trace elements, such as As, Cd, Cr, Sb, Se, and Pb, are highly toxic to the environment. Moreover, the irrigation of agricultural soils with water with high concentrations of TEs does not only result in the transfer of those elements from the water to the soil, but also increases the bioavailability of these analytes for trees and plants [11].

The solubility and bioavailability of the elements in the plants may be affected by physicochemical parameters such as pH and EC [30].

The salinity in the soil is mainly measured by Electrical Conductivity (EC). The EC in the soil showed values from 0.17 to 1.40 mS/cm. In the Las Bebas orchard, the EC value was 1.40 mS/cm, which can be attributed to the continuous application of treated irrigation water on the soil, which can increase the EC. In some cases, soil with high EC resulting from a high concentration of sodium can become toxic to plants (Table 3), generally caused by a poor structure and bad drainage in the soil. The harvest at Las Bebas was considered unsuccessful with no growth for the apples. The high levels of precipitation can flush soluble salts out of the soil and reduce EC; the Rancho Verde, Cristina, and La Esperanza orchards were sampled in 2019, which was a rainy year in the region. On the other hand, the Solis, La Mesa, and Las Bebas orchards were sampled in 2020, a drought year. However, the EC from 0 to 2 mS/cm is a non-saline class [31].

Contrarily, the EC in irrigation water showed values of 0.173 to 0.380 mS/cm; it did not exceed the degree of restriction of 0.7 mS/cm [32].

The concentrations of As, Cu, Cd, and Cr in the soil were above the world soil average of 4.7, 14, 1.1, and 0.3 mg·kg^−1^ [32]. In the Cristina orchard, Cu concentration was 408.6 mg·kg^−1^, and the phytotoxic levels of Cu in the soil were from 36 to 698 mg·kg^−1^, which is attributed to the regular long-term utilization of Cu-based sprays. On the other hand, the results in Table 3 show high concentrations of Na from 1047 to 1571 mg·kg^−1^ in three orchards. An excess of Na is frequently assumed to be largely responsible for the reductions in growth and yield under salinity, which is attributed to limited rainfall [33,34].

The concentrations of Ca, Mg, and P in the samples of soil in the orchards with low levels of precipitation in Solis, La Mesa, and Las Bebas were lower than in the Cristina, Rancho Verde, and La Esperanza orchards, which can be attributed to the difference in the precipitation levels and EC between the years of sampling.

The concentrations of major and TEs in the irrigation water that were analyzed in this research were below the recommended maximum concentration for irrigation water (Table 2) by CONAGUA, which is the National Water Commission in Mexico, which is an administrative organization of the Ministry of Environment and Natural Resources, created in 1989, whose responsibility is to manage, regulate, control, and protect national waters in Mexico.

### 4.2. Element Characterization in Pyrus malus

The elements in the environment can enter the food chain through different pathways delivering toxic effects on health [12]. Accordingly, it is fundamental to understand the incidence of TEs in plants, above all, those intended for eating, and to realize if there are differences in patterns of accumulation by different plant varieties to give useful information to consumers.

The measured concentration of As in the Cristina, Rancho Verde, and La Esperanza orchards ranged from 0.18 to 0.34 mg·kg^−1^ in apples and ranged from 0.93 to 1.83 mg·kg^−1^ in the peel. These values exceeded the maximum allowable level of As for fruits 0.05 mg·kg^−1^. In the same way, in the literature, a similar study in Serbia reported a mean concentration of As of 1.25 mg·kg^−1^ in apples [35]. This value is within the range obtained in this study for the peel. One study of honey in Armenia reported that As ranged from 0.005 to 0.048 mg·kg^−1^ [10] and was close to the values detected in China in maize 0.0428 mg·kg^−1^ [7]. The As values in apples reported in the present study are higher than the majority of As values in fruits [7,10,35,36]. In the orchards with low levels of precipitation, Solis, La Mesa, and Las Bebas, arsenic was not detected, which can be attributed to the difference in precipitation between the years of sampling (2020 was a drought year).

The concentrations of Cd in the present study ranged from 0.11 to 0.14 mg·kg^−1^ in apple samples; on the other hand, in peel samples, Cd ranged from 0.04 to 0.25 mg·kg^−1^. All the orchards exceeded the maximum allowable level of Cd for fruits 0.05 mg·kg^−1^. On the same hand, in the literature, one study of apples in Italy that reported Cd ranging from 0.011 to 0.019 mg·kg^−1^ [37] can be found, which is below the results obtained in this study. In comparison to other studies, Cd in maize was reported to range from 0.0025 to 0.2134 mg·kg^−1^, similar to the range in results obtained [7]. Elevated concentrations of Cd can be attributed to the earlier application of agrochemicals that contained this analyte; nevertheless, the utilization of that chemical is no longer used. Cd also has a high solubility in water as well as relatively high mobility in the soil–plant system.

The mean Cr concentrations in apples were evaluated in the range from 0.08 to 0.86 mg·kg^−1^; for peels, the concentration of Cr ranged from 0.4 to 3.2 mg·kg^−1^. It did not exceed the maximum allowable level for fruits 1 mg·kg^−1^. However, the Rancho Verde, La Esperanza, and Las Bebas orchards exceeded that maximum allowable level. Furthermore, a study in China reported Cr ranged from 0.1206 to 0.6142 mg·kg^−1^ in maize [7].

These results indicate that As, Cd, and Cr pollution is a serious threat to apple safety and poses a high potential risk to human health.

Concentrations of Sb found in the peel in the present study ranged from 1.1 to 2.5 mg·kg^−1^; Sb in the apple was not detected. In the literature, the values reported for this element are in carrots (0.02–0.03 mg·kg^−1^), red beets (0.02–0.09 mg·kg^−1^), onions (0.02–0.03 mg·kg^−1^), and potatoes (0.02–0.02 mg·kg^−1^ [38]. Other studies in plants show that antimony was especially translocated in leaves; on the other hand, storage organs, seeds, and fruits had lower concentration values. However, in the present study, the capacity to accumulate Sb in apples is also in the peel, although in a lower concentration.

#### Concentrations of Major and Toxic Elements in Apple Leaves

The concentrations of As, Cd, and Cr in the leaves were found in a range between 0.61 and 4.94 mg·kg^−1^; 0.50 and 1.17 mg·kg^−1^; 0.68 and 2.71 mg·kg^−1^; however, in the La Mesa, Solis, and Las Bebas orchards, these analytes were not detected, which can be attributed to the difference in precipitation between the years of sampling, 2020 being a year of drought or water deficit. Although considerable studies on TEs and water deficit as individual stresses have been conducted, studies with simultaneous exposure to these two stresses in the literature are very few [39]; however, the results also showed that the transfer of trace elements is decreased in the plant, due to water deficit, water being one of the most important factors for plant development.

The concentration of Cu in the leaves was found to range between 3.53 and 15.97 mg·kg^−1^. In the literature, the mean reported for this analyte in apple leaves is 16.320 mg·kg^−1^ and in peach leaves, it is 33.0 mg·kg^−1^ [35].

On the other hand, the concentration of Na ranged from 182 to 842 mg·kg^−1^; however, in the Las Bebas orchard, the mean concentration of this analyte was 2403 mg·kg^−1^; this high value can be attributed to a drought or water deficit at that orchard.

### 4.3. Health Risk Assessment

#### 4.3.1. Non-Carcinogenic Risk Assessment

The order of the mean HQ value was Cd > As > Zn > Cr. When the HQ is equal to or higher than one, the element represents a health risk. The HQ value in As, Cd, Cr, and Zn in the most contaminated orchards Cristina, Rancho Verde, and La Esperanza were lower than one; however, it is important to note that the prolonged ingestion of these trace elements may pose a health risk, due to their high concentration in apples.

In the present study, the HQ for As in apples was in a range of 0.167 to 0.315, Cd of 0.306 to 0.389, Cr of 0.001 to 0.002, and Zn of 0.051 to 0.149. In contrast, research in China found HQ values for As of 1.09, Cd of 0.3, Cr of 0.8 in maize, respectively [7]. On the other hand, a study in Armenia found a maximum value of As in honey of 0.0432 [10]. Finally, fruits and vegetables in Bangladesh found HQs for Cr of 0.0004, 0.002, and 0.006 for banana, bean, and tomato, respectively [40].

#### 4.3.2. Carcinogenic Risk Assessment

Depending on the time of exposure and intake dose, an element can represent a potential carcinogenic risk for a person. According to the EPA, there are three cancer risk levels to assess the carcinogenic risk of an element. The first one is the CR < 10^−6^, which is the safe limit indicating that there is no risk; the second one is if CR is in the range of 10^−6^ to 10^−4^, which indicates an acceptable cancer risk; CR > 10^−4^ is the threshold limit and it is considered unacceptable since it can represent a health risk for people [41]. The CRs in the most contaminated orchards were: As 1.4 × 10^−3^, Cd 1.3 × 10^−4^, and Cr 9.58 × 10^−4^. These results indicate that there is a possible carcinogenic risk caused by these elements if there is prolonged ingestion of these apples. On the same hand, this result indicates that these TEs should be of preoccupation because they exceed the safety values of the carcinogenic risk. This finding indicates that even without other plausible rich sources of those TEs (rice, drinking water, fruits, vegetables, etc.), apple consumption itself is already a possible carcinogenic risk for the apples produced in the studied regions in Chihuahua.

Another study on maize in China found CR for As (1.17 × 10^−4^), Cd (4.93 × 10^−4^), and Cr (2.74 × 10^−4^) [7]. On the same hand, a study of honey in Armenia found CR of Cd ranged from 4.79 × 10^−8^ to 2.48 × 10^−7^ and As from 1.58 × 10^−6^ to 1.51 × 10^−5^ [10]. Finally, a study of tomato in India found CR of As ranged from (0.0006–6.963), of Cd (0.00042–0.04962), and of Cr (0.0087–0.041), respectively [41].

## 5. Conclusions

The present research presented the concentration of the major and toxic heavy metals in agricultural soils, irrigation water, and apple trees collected from the agricultural area in Chihuahua, Mexico. The study results highlight the importance of investigating exposure to trace elements through apple consumption. In the studied regions, apple is consumed in high amounts and this has an important contribution to the intake of investigated trace elements. High levels of As, Cd, and Cr were found in the apple samples; however, As and Cd were translocated in the apples in a year with high precipitation and only Cr can be absorbed by the plant in drought conditions. On the other hand, the HQ for As, Cd, and Cr indicate a potential health risk in the short term. However, it is important to note that the prolonged ingestion of As, Cd, and Cr may pose a health risk. Meanwhile, carcinogenic risk values of As, Cd, and Cr exceed the threshold limit and are considered unacceptable (CR > 10^−4^) in the studied region which shall warrant concern.

It can be concluded that there is a difference in the translocation of trace elements in a year with rain and a year with drought or water deficit. Drought is an extended threat to crops. Ultimately, the TEs could be transferred from the soil to the apple tree and increase the availability of toxic elements to the apple trees. Therefore, it is suggested that the Mexican authorities establish health risks assessments and legislation with maximum allowable concentrations for these toxic elements in fruits and vegetables to monitor.

## Figures and Tables

**Figure 1 ijerph-19-12032-f001:**
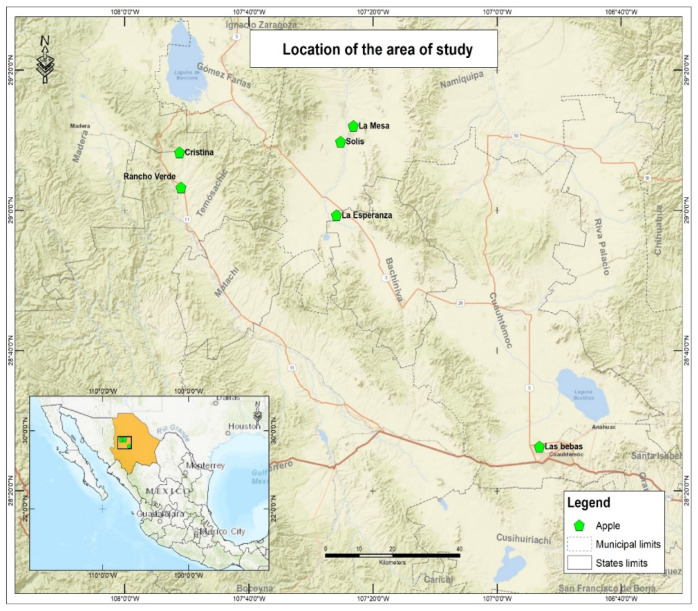
Area of study.

**Figure 2 ijerph-19-12032-f002:**
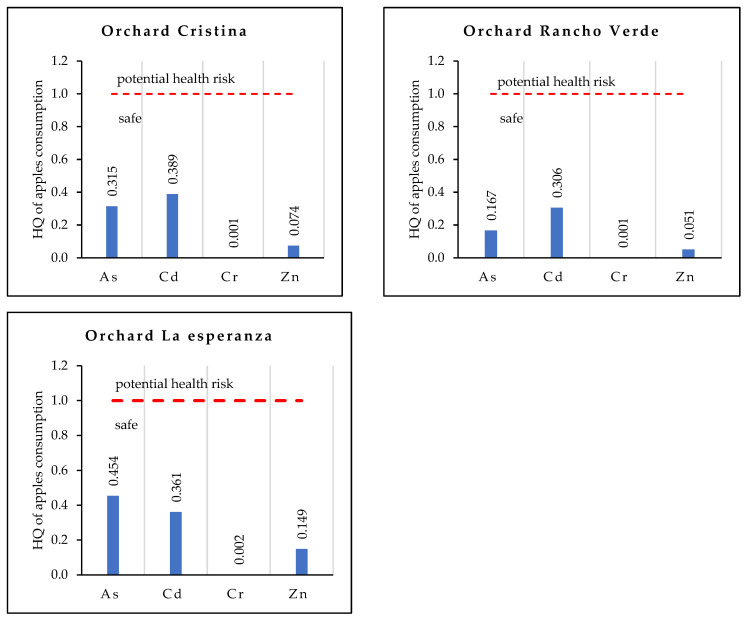
Hazard Quotient (HQ) of trace elements of the most contaminated orchards.

**Figure 3 ijerph-19-12032-f003:**
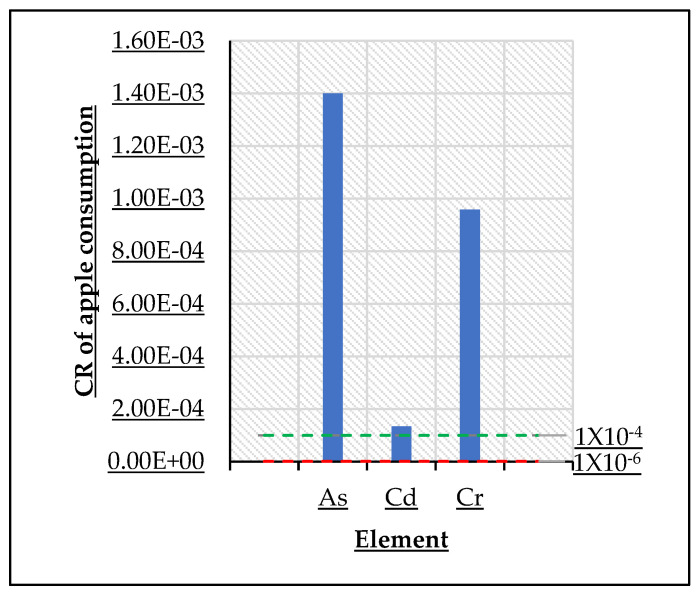
CR of apple consumption.

**Table 1 ijerph-19-12032-t001:** Chemical parameters measured in situ in soils and irrigation water of the orchards.

Orchard	Variety	Latitude	Longitude	pH		EC (mS/cm)	
				Water	Soil	Water	Soil
Rancho Verde	Golden Delicious	29°3′15.58″	107°50′55.35″	7.24	6.95	0.173	0.168
Cristina	Golden Delicious	29°8′16.89″	107°51′9.64″	7.31	6.78	0.226	0.231
La Esperanza	Golden Delicious	28°59′19.59″	107°25′51.30″	7.51	7.6	0.377	0.178
Solis	Golden Delicious	29°9′48.90″	107°25′11.54″	8.14	6.92	0.244	0.269
La Mesa	Golden Delicious	29°12′3.69″	107°23′7.82″	8.07	7.94	0.343	0.385
Las Bebas	Golden Delicious	28°26′17.8″	106°53′7.00″	7.61	5.54	0.380	1.406

**Table 2 ijerph-19-12032-t002:** Concentrations of elements in irrigation water. Data are reported in mg·L^−1^ as mean ± SD.

Element	Water					Ala	Alb
	Cristina and Rancho Verde	La Esperanza	Solis	La Mesa	Las Bebas		
As	0.003 ± 0.0002	0.007 ± 0.002	0.009 ± 0.002	0.030 ± 0.0005	0.030 ± 0.0037	0.400	0.100
Ca	14.25 ± 2.66	20.77 ± 1.20	23.4 ± 0.70	27.9 ± 0.36	40.6 ± 0.35	-	-
Cr	0.015 ± 0.0029	0.001 ± 0.0002	0.006 ± 0.0004	0.008 ± 0.001	0.006 ± 0.0015	1	0.1
Cu	0.004 ± 0.002	0.004 ± 0.001	0.027 ± 0.002	0.033 ± 0.002	0.036 ± 0.006	-	-
Fe	5.2 ± 0.8	0.50 ± 0.2	0.09 ± 0.007	0.06 ±0.001	0.10 ± 0.003	-	-
K	14.7 ± 2.2	2.5 ± 0.09	12.7 ± 0.9	18.1 ± 6.2	15.7 ± 0.29	-	-
Mg	4.3 ± 0.60	2.4 ± 0.23	3.4 ± 0.06	2.5 ± 0.05	2.2 ± 0.03	-	-
Mn	0.18 ± 0.003	0.04 ± 0.009	0.004 ± 0.00001	0.002 ± 0.0006	0.02 ± 0.0007	-	0.2
Na	10.0 ± 0.33	40.6 ± 2.10	15.3 ± 0.2	40.3 ± 1.2	129.5 ± 2.4	-	-
P	0.66 ± 0.19	0.04 ± 0.01	1.0 ± 0.03	1.1 ± 0.03	1.9 ± 0.01	-	-
S	0.9 ± 0.15	4.3 ± 0.33	2.6 ± 0.07	5.5 ± 0.13	35.6 ± 0.39	-	-
Sb	<LOD	<LOD	<LOD	<LOD	28 ± 6.4	-	-
Zn	0.024 ± 0.002	0.004 ± 0.001	0.0341 ± 0.015	0.025 ± 0.002	0.064 ± 0.002	20	2

Ala = Allowed limit by CONAGUA. Alb = Allowed limit by FAO. <LOD = Result lower than the detection limits of ICP-OES—there is no allowed limit established by agencies. Note: Data are the mean of n = 7. Cd, Pb, and Se were not found in irrigation water.

**Table 3 ijerph-19-12032-t003:** Concentrations of elements in the soil. Data are reported in mg·kg^−1^ as mean ± SD.

Element	Soil					
	Cristina	Rancho Verde	La Esperanza	Solis	La Mesa	Las Bebas
As	24.42 ± 3.3	22.25 ± 2.4	18.79 ± 1	<LOD	<LOD	<LOD
Ca	11,505 ± 927	14,256 ± 730	39,777 ± 4695	14,811 ± 560	17,301 ± 3433	8819 ± 73
Cd	1.4 ± 0.0002	1.48 ± 0.0001	4.1 ± 0.0001	<LOD	<LOD	<LOD
Cr	11.5 ± 1.1	13.5 ± 1.3	12.3 ± 2.9	8.6 ± 1.2	5.4 ± 1.2	12.9 ± 1.5
Cu	408.6 ± 148.3	51.4 ± 12.3	52.4 ± 4.9	4.0 ± 0.8	5.2 ± 2.0	5.8 ± 0.4
Fe	12,283 ± 1603	13,928 ± 1538	14,807 ± 4027	80,380 ± 10,495	63,087 ± 7510	11,2305 ± 12,446
K	2407 ± 261	2771 ± 360	1750 ± 231	11,310 ± 1276	12,805 ± 3040	15,393 ± 2853
Mg	2331 ± 355	2289 ± 156	1444 ± 237	207 ± 5.7	203 ± 8.2	196 ± 10.1
Mn	533 ± 36	426 ± 45	392 ± 22	185 ± 41	125 ± 7	143 ± 21
Na	199 ± 19	177 ± 29	249 ± 45	1571 ± 224	1047 ± 137	1352 ± 253
P	1226 ± 105	1892 ± 194	419 ± 66	202 ± 42	168 ± 30	258 ± 13
Pb	19.8 ± 1.8	20.8 ± 3.7	15.0 ± 1.7	5.9 ± 1.1	5.1 ± 0.3	8.8 ± 0.5
S	484 ± 67	343 ± 83	172 ± 8	130 ± 31	72 ± 5	125 ± 47
Zn	61.4 ± 14	73.6 ± 14	44.3 ± 14	19.6 ± 6	20.2 ± 3	17.8 ± 2

Note: Data are the mean of n = 7. Sb and Se were not found in soil.

**Table 4 ijerph-19-12032-t004:** The concentration of major and trace elements in apple and peel. Data are reported in mg·kg^−1^, as mean ± SD.

Element	Apple							Peel					
	Cristina	Rancho Verde	La Esperanza	Solis	La Mesa	Las Bebas	LPa	Cristina	Rancho Verde	La Esperanza	La Mesa	Solis	Las Bebas
As	0.34 ± 0.14	0.18 ± 0.03	0.49 ± 0.14	<LOD	<LOD	<LOD	0.05	0.93 ± 0.11	1.61 ± 0.69	1.83 ± 0.57	<LOD	<LOD	<LOD
Ca	223 ± 21.9	214 ± 14.7	317 ± 53.3	152 ± 32.5	551 ± 86.6	161 ± 20.8	-	1232 ± 89	1085 ± 155	889 ± 177	312 ± 71	378 ± 113	315 ± 51
Cd	0.14 ± 0.02	0.11 ± 0.01	0.13 ± 0.02	<LOD	<LOD	<LOD	0.05	0.02 ± 0.01	0.25 ± 0.02	0.24 ± 0.02	<LOD	<LOD	0.04 ± 0.03
Cr	0.60 ± 0.17	0.61 ± 0.12	0.86 ± 0.28	0.08 ± 0.02	0.21 ± 0.05	0.77 ± 0.11	1	0.60 ± 0.08	1.10 ± 0.39	3.20 ± 1.13	0.40 ± 0.08	0.54 ± 0.08	1.70 ± 0.53
Cu	1.1 ± 0.28	1.9 ± 0.28	2.0 ± 0.22	1.9 ± 0.40	3.0 ± 0.34	6.1 ± 1.20	4.5	1.7 ± 0.43	2.1 ± 0.25	2.0 ± 0.36	6.0 ± 1.05	2.0 ± 0.70	1.2 ± 0.31
Fe	6.8 ± 0.96	3.2 ± 0.39	8.6 ± 1.50	6.5 ± 0.27	10.6 ± 0.85	34.2 ± 8.14	-	11.9 ± 2.83	13.8 ± 4.34	31.3 ± 5.34	16.6 ± 2.72	16.0 ± 2.85	20 ± 6.13
K	9428 ± 2664	16,381 ± 2499	10,322 ± 998	7311 ± 685	8055 ± 1040	7401 ± 363	-	12,983 ± 309	15,043 ± 4414	11,241 ± 2589	7294 ± 1170	7609 ± 655	5674 ± 532
Mg	258 ± 23	308 ± 26	338 ± 19	168 ± 10	194 ± 13	199 ± 16	-	645 ± 70	721 ± 61	864 ± 94	272 ± 22	321 ± 15	276 ± 22
Mn	1.16 ± 0.14	1.59 ± 0.21	2.17 ± 0.27	1.30 ± 0.24	1.94 ± 0.31	3.90 ± 0.57	-	5.0 ± 0.90	6.8 ± 1.04	8.2 ± 1.11	4.3 ± 1.15	3.3 ± 0.57	2.3 ± 0.33
Na	184 ± 14.4	204 ± 38.0	142 ± 18.9	61 ± 4.1	244 ± 33.3	65 ± 18	-	359 ± 62.0	245 ± 26.7	167 ± 6.2	98 ± 24.1	39 ± 11.7	58 ± 14.1
P	594 ± 37.6	651 ± 40.6	689 ± 54.0	528 ± 55.2	531 ± 92.5	412 ± 65.7	-	1529 ± 218	1566 ± 274	1796 ± 210	424 ± 119	793 ± 127	307 ± 28
S	177 ± 29	191 ± 23	326 ± 9	101 ± 14	229 ± 30	139 ± 47	-	396 ± 50	417 ± 35	472 ± 42	221 ± 43	166 ± 30	184 ± 54
Sb	<LOD	<LOD	<LOD	<LOD	<LOD	<LOD	-	<LOD	<LOD	<LOD	1.6 ± 0.56	1.1 ± 0.26	2.5 ± 0.73
Zn	8.0 ± 1.2	5.5 ± 0.9	16.0 ± 2.7	1.7 ± 0.1	4.8 ± 0.08	5.0 ± 0.6	-	4.0 ± 0.9	2.7 ± 1.4	8.0 ± 1.7	10.5 ± 2.6	8.3 ± 1.3	7.0 ± 0.7

Note: Data are the mean of n = 7. Pb and Se were not detected. LPa = Permissible limit (FAO/WHO, 2011).

**Table 5 ijerph-19-12032-t005:** Concentration of major and trace elements in leaves and Reference Material Analysis. Data are reported in mg·kg^−1^, as mean ± SD.

Apple Leaves							NIST SRM 1515 Apple Leaves			
Orchard	Cristina	Rancho Verde	La Esperanza	La Mesa	Solis	Las Bebas	Certified Value ± U	Measured Value ± U	Mean Recovery (%) (*n* = 10)	RSD (%) (*n* = 10)
As	0.61 ± 0.17	4.94 ± 0.10	4.81 ± 0.20	<LOD	<LOD	<LOD	N.A.	0.012 ± 0.003	106.32	7.2
Ca	15,080 ± 227	12,542 ± 958	16,875 ± 1600	19,116 ± 1812	16,346 ± 150	13,028 ± 1016	15,250 ± 100	15,301 ± 74	100.33	2.9
Cd	<LOD	1.17 ± 0.09	0.50 ± 0.02	<LOD	<LOD	<LOD	0.0132 ± 0.0015	0.0135 ± 0.0012	102.27	6.2
Cr	1.84 ± 0.16	1.83 ± 0.22	2.71 ± 0.54	0.84 ± 0.10	0.69 ± 0.11	0.68 ± 0.12	N.A.	0.42 ± 0.03	103.25	4.6
Cu	3.53 ± 0.8	4.28 ± 0.9	4.90 ± 0.4	15.97 ± 2.2	7.92 ± 1.6	15.70 ± 1.2	5.69 ± 0.13	5.57 ± 0.15	97.89	3.1
Fe	78.3 ± 6.7	139.6 ± 20.3	165.7 ± 28.2	192.4 ± 20.9	90.8 ± 13.4	110.3 ± 2.7	82.7 ± 2.6	80.5 ± 3.2	97.34	3.2
K	13,646 ± 2157	23,672 ± 4004	15,121 ± 1186	18,392 ± 1742	19,721 ± 1708	12,631 ± 1284	16,080 ± 210	17,010 ± 178	105.78	4.1
Mg	2125 ± 367	2475 ± 381	3283 ± 188	559 ± 13	550 ± 23	583 ± 10	2710 ± 120	2801 ± 154	103.36	2.3
Mn	14.08 ± 2.6	25.95 ± 4.9	42.69 ± 4.0	59.17 ± 4.3	19.81 ± 3.3	45.34 ± 2.6	54.1 ± 1.1	55.1 ± 1.4	101.85	2.9
Na	292 ± 38	198 ± 22	182 ± 17	842 ± 145	283 ± 27	2403 ± 282	24.4 ± 2.1	24.2 ± 2.5	99.18	2.8
P	3886 ± 493	3459 ± 115	2981 ± 373	2430 ± 9	3010 ± 252	1335 ± 134	1593 ± 68	1550 ± 80	97.30	4.1
Pb	1.61 ± 0.1	<LOD	<LOD	<LOD	<LOD	<LOD	0.470 ± 0.024	0.490 ± 0.030	104.26	5.2
S	1259 ± 130	958 ± 53	1262 ± 83	1569 ± 116	1213 ± 136	1672 ± 183	N.A.	1042 ± 72	98.23	3.6
Zn	19.2 ± 3.0	11.5 ± 1.8	25.9 ± 4.5	118.2 ± 12.2	7.9 ± 1.6	31.6 ± 4.1	12.45 ± 0.43	12.8 ± 0.65	102.81	4.2

Note: Data are the mean of n = 7. Sb and Se were not detected.

**Table 6 ijerph-19-12032-t006:** CR of apple consumption.

Element	CR of Apple
As	1.40 × 10^−3^
Cd	1.34 × 10^−4^
Cr	9.58 × 10^−4^

## Data Availability

Not applicable.

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
