# Peer review of "Dynamics of Major and Trace Elements in Water–Soil–Tree Interaction: Translocation in Pyrus malus in Chihuahua, Mexico Using ICP-OES and Its Health Risk Implications"

_ijerph, 2022, doi:10.3390/ijerph191912032_

Round 1
Reviewer 1 Report (Previous Reviewer 2)
The manuscript is well-conducted and the authors have explained correctly the material and methods. The results and discussion are explained and compared with other authors.
Author Response
Thanks. The spell check has been performed by a native english speaker and the changes have been applied.
Reviewer 2 Report (New Reviewer)
The manuscript entitled “Dynamics of major and trace elements in water-soil-tree interaction: Translocation in Pyrus malus in Chihuahua, Mexico using ICP-OES. Health Risk Implications.” focused on the concentrations of elements, such as As, Ca, Cd, Cu, Cr, Fe, K, Mg, Mn, P, Pb, S, Sb, Se and Zn in the soil, leaves, peel, walnut, apple, and irrigation water.
I think that the paper can be accepted for publication after a few minor revisions according to my following comments.
1. It is not clear from the paper how the elements here measured were chosen.
2. Is it possible to add a fragment of the text about the sources of heavy metal emissions in the study area?
3. Has rainwater been determined?
4. Line 263 Is it correct unit? 0.7 mS/m
Author Response
Thanks for your comments. Please find attached the answers to the questions.

This manuscript is a resubmission of an earlier submission. The following is a list of the peer review reports and author responses from that submission.
Round 1
Reviewer 1 Report
This is a great idea for a research project, and some of your results show reason for concern about contaminants levels in apples. The following points would need to be addressed for this article to be publishable:
1) When you state that no similar studies have been published "in the apple farming region to the west in the state of Chihuahua Mexico," the reader wonders what other comparable research has been done in other parts of Chihuahua? in Mexico? Elsewhere in the world? It appears there is a gap here in the literature review.
2) Abstract says the samples were collected in 2020 and 2021, the text says 2019 and 2020. This is a serious inconsistency.
3) Far more information is needed to describe your sample collection methods. What did you do in designing your sampling protocol, for each of the sample types, to ensure that your samples are representative of that orchard? What did you do to ensure your sample bottles and containers were not contaminated with trace elements? What did you use to peel apples, to ensure your peeler was not adding trace metals to the samples?
4) Additional information is needed on your QA/QC. What were the calibration curve ranges and LOD for each element? Were blanks and spikes run for each element? Table 5 gives useful QA/QC info for apple leaves. Please provide the comparable QA/QC data for the other sample types.
5) Correct the mistakes in Equation 3 and lines 147-150.
6) Line 148, it appears RfDs are in units of ug/kg, while your concentrations are in mg/kg (line 137). Where does your equation correct for that?
7) Please list each RfD value used and provide a citation for each.
8) Line 158, provide a citation for the Slope Factors for As, Cd and Cr.
9) Lines 158-159, you cite 1 x 10-6 as the acceptable level for human health, and later in the article cite 1 x 10-4, and in another location, mention both levels. If you wish to use and compare data against both CR values, state so up front (line 158).
10) Line 181. Define CONAGUA for readers who may not be familiar with it.
11) Table 4. Label the data so reader knows which data are apple and which are peel.
12) Line 234, please provide citation for the 200g per day consumption level.
13) Figure 2: Are these data from all of the orchards combined? From just the most contaminated orchard? Please show data separated out by site, or at least show averages and standard deviation among orchards. Is this based on consumption of just apple, or apple + peel?
14) Line 241, incorrect use of scientific notation, and no explanation as to why E-04 rather than E-06 as stated above.
15) The English throughout needs substantial editing for the meaning to be clear and the sentences to be grammatically correct.
16) Section 4.2. There are some useful citations of contaminant levels in various vegetable crops and (puzzlingly) in honey. These citation should be in the lit review section, with an explanation as to how levels in vegetables are relevant to levels in apples.
17) Lines 293-294: "The values in apples reported in the present study are higher than the majority of As values in fruits." - this is a key statement, made without any citation, and it absolutely needs multiple citations.
18) Lines 308-309: "This finding indicates that apples have a higher capacity to accumulate As, Cd and Cr." No - it does not indicate that at all. The "finding" referenced is just about Cr, and more importantly, the higher levels could just be because the soil and/or water were more polluted. You cannot conclude anything about "capacity to accumulate."
19) Lines 338-339: The conclusion stated here is contradictory to the conclusion stated in lines 344-345, and is contradictory to the data presented in Figure 2.
20) Sections 4.3.1 and 4.3.2: Citations of studies done on honey, maize, beans, etc. are not sufficiently framed to explain their relevance to the conclusions presented.
Reviewer 2 Report
The manuscript is not well-conducted. The authors does not explain enough the importance of the Pyrus malus in this region. What is about the pesticides used? What is about other possible metal sources?
Why authors determine macro-elements? Is there any source of Ca, K, Mg or Na near to the crops?
In this article they have tried to cover many aspects (carcinogenic risk studies...). From my point of view, the authors should focus on the study of the transfer of toxic or potentially toxic metals between soil - water - tree.
Is the type of soil unknown, the element content determined from where it comes from? In the abstract they say that samples have been taken since 2020, but in material and methods it indicates that they started to be taken in 2019.
The authors should correctly define the material and methods used. The statistical study is poorly explained, state better what has been done and justify why.
Do not repeat mg/kg in the tables and in the titles, indicate only in one.
There is no reference material for the study of metals in water and soil? Only indicated for apple leaves.
Do not use values with exponents, as they are not used throughout the manuscript. They must follow the same model.
The authors comment on the carcinogenic risk, but As has been specified? Do we know what type of As it is? The same for Cr.
The authors talk about Sb, but they have not measured it.
The manuscript needs a major revision.